# PETRA: PARALLEL END-TO-END TRAINING OF REVERSIBLE ARCHITECTURES

**Stéphane Rivaud**[1,5] **Louis Fournier**[1] **Thomas Pumir**[6]
**Eugene Belilovsky**[3,4] **Mickael Eickenberg**[2] **Edouard Oyallon**[2,*]

[1]ISIR – Sorbonne Université, Paris   [2]Flatiron Institute, New York

[3]Mila, Montréal   [4]Concordia Université, Montréal

[5]LISN – Université Paris-Saclay, CNRS, Inria, Orsay   [6]Helm.ai, San Francisco

`stephane.a.rivaud@inria.fr`

## ABSTRACT

Reversible architectures have been shown to be capable of performing on par with their non-reversible architectures, being applied in deep learning for memory savings and generative modeling. In this work, we show how reversible architectures can solve challenges in parallelizing deep model training. We introduce PETRA, a novel alternative to backpropagation for parallelizing gradient computations. PETRA facilitates effective model parallelism by enabling stages (i.e., a set of layers) to compute independently on different devices, while only needing to communicate activations and gradients between each other. By decoupling the forward and backward passes and keeping a single updated version of the parameters, the need for weight stashing is also removed. We develop a custom autograd-like training framework for PETRA, and we demonstrate its effectiveness on CIFAR-10, ImageNet32, and ImageNet, achieving competitive accuracies comparable to backpropagation using ResNet-18, ResNet-34, and ResNet-50 models.

## 1 INTRODUCTION

First-order methods using stochastic gradients computed via backpropagation on mini-batches are the de-facto standard for computing parameter updates in Deep Neural Networks (LeCun et al., 2015). As datasets and models continue to grow (Alabdulmohsin et al., 2022) there is an urgent need for memory-efficient and scalable parallelization of deep learning training across multiple workers. Data parallelism via mini-batches (LeCun et al., 2015) has been widely adopted in deep learning frameworks (Li et al., 2020). This approach computes gradients across model replicas distributed among workers, yet it requires frequent synchronization to aggregate gradients, leading to high communication costs, as well as substantial memory redundancy. Furthermore, with the increasing size and scale of models exceeding that of the growth of on-device memory, the forward and backward passes now often exceed a single device's memory capacity (Ren et al., 2021). To further address these issues, methods have attempted to mitigate this memory overhead and to parallelize the sequential backpropagation steps themselves across devices, while computing exact gradients. Techniques like optimizer sharding (Rajbhandari et al., 2020), tensor parallelism (Shoeybi et al., 2019), activation checkpointing (Chen et al., 2016), or pipelining (Huang et al., 2019), have been deployed individually or combined, leading for instance to the development of 3D parallelism (Smith et al., 2022), a popular methodology which improves the efficiency of the backpropagation implementation. On the other hand, the fundamental inefficiency underlying the parallelization of backpropagation has not been addressed by these methods.

However, the use of exact gradient restricts algorithmic choices and parallel implementations, as highlighted by Jaderberg et al. (2017). For instance, backpropagation is *backward locked*: the inputs of each layer must be propagated through the network and preserved until an error signal is

---

[*]During a one year leave - now back at CNRS, Sorbonne University.

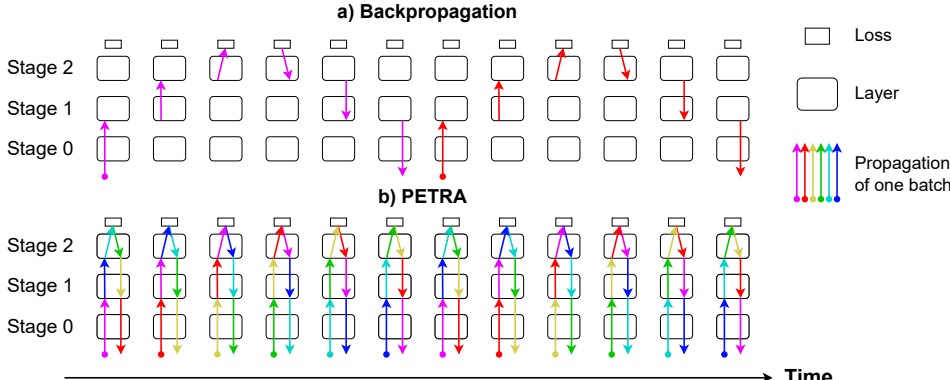

Figure 1: **Comparison of PETRA with standard backpropagation.** This approach splits the stages of a model and decouples their forward and backward passes, resulting in a sixfold increase in parallelization speed in this example.

retropropagated to the layer of origin. This requirement enforces a synchronous dependency among subsequent layers and requires them to systematically store intermediary activations, potentially impeding overall resource efficiency as workers must wait for each other to continue their computations and release memory used for activations. To unlock the potential of backpropagation, inexact backpropagation procedures have been proposed. These procedures are generally conceptualized within the context of model parallelism, where a neural network is split into stages that can process their activations in parallel, potentially on multiple devices. For example, some methods use outdated parameters or activations, such as double-buffered pipelining (Harlap et al., 2018) or delayed gradient approaches (Zhuang et al., 2021b). However, these methods introduce significant memory overhead due to the use of ad hoc buffers for activations, parameters, or both. Following an opposite direction, local learning methods (Nøkland & Eidnes, 2019; Belilovsky et al., 2020), which estimate inexact gradients via a local auxiliary neural network, pave the way to parallel gradient computations but often lead to unrecoverable performance drops (Fournier et al., 2023). This underscores the need for a robust alternative to backpropagation, with limited memory overhead.

In this work, we introduce PETRA (Parallel End-to-End Training with Reversible Architectures), a novel method designed to parallelize gradient computations within reversible architectures with minimal computational overhead. Reversible architectures are an ideal candidate for this task, as they can significantly reduce memory overhead during standard backpropagation with limited communication costs. Furthermore, reversibility is a minor requirement, as many studies have demonstrated that standard architectures can be adapted into reversible ones without any performance drops (Gomez et al., 2017; Jacobsen et al., 2018b; Mangalam et al., 2022; Kitaev et al., 2020). By allowing parameters to evolve in parallel and by computing an approximate inversion during backward, we propose an effective alternative to backpropagation which allows high model parallelism with a constant communication overhead and **no additional parameter or activation buffers**. In fact, for a constant increase in communication overhead, PETRA achieves a linear speedup compared to standard backpropagation with respect to the number $J$ of stages the network is split into. We illustrate our approach in Fig. 1, by contrasting the evolution of PETRA with a standard backpropagation pass.

**Contributions.** Our contributions are as follows: **(1)** We introduce PETRA, a streamlined approach for parallelizing the training of reversible architectures. This method leverages a delayed, approximate inversion of activations during the backward pass, allowing for enhanced computational efficiency. **(2)** Our technique significantly reduces memory overhead by minimizing the necessity to store extensive computational graphs. **(3)** It enables the parallelization of forward and backward pass computations across multiple devices, effectively distributing the workload and reducing training time. **(4)** We validate the efficacy of PETRA through rigorous testing on benchmark datasets such as CIFAR-10, ImageNet-32, and ImageNet, where it demonstrates robust performance with minimal impact on accuracy. **(5)** We observe a significant empirical throughput

increase when using PETRA. **(6)** Additionally, we provide a flexible reimplementation of the autograd system in PyTorch, specifically tailored for our experimental setup, which is available at `https://github.com/stephane-rivaud/PETRA`.

## 2 RELATED WORK

**Reversible architectures.** Reversible DNNs are composed of layers that are invertible, meaning that the input of a layer can be computed from its output. This approach allows to avoid the need to store intermediary activations during the forward pass by reconstructing them progressively during the backward pass (Gomez et al., 2017), at the cost of an extra computation per layer. Invertible networks further improve this method by removing dimensionality reduction steps such as downsamplings, making the networks fully invertible (Jacobsen et al., 2018a). Reversibility is not restricted to a type of architecture or tasks and has been extensively used for generative models (Dinh et al., 2014), for ResNets (Gomez et al., 2017), and Transformers (Mangalam et al., 2022). However, as far as we know, reversible architectures have never been used to enhance parallelization capabilities.

**Alternatives to backpropagation.** Multiple alternatives to backpropagation have been proposed previously to improve over its computational efficiency. For instance, DNI (Jaderberg et al., 2017) is the first to mention the backpropagation inefficiency and its inherent synchronization locks. However, they address those locks with a method non-competitive with simple baselines. Local (or greedy) learning (Nøkland & Eidnes, 2019; Belilovsky et al., 2019) propose to use layerwise losses to decouple the training of layers, allowing them to train in parallel (Belilovsky et al., 2021). Local learning in videos (Malinowski et al., 2021) notably uses the similarity between successive temporal features to remove buffer memory. However, the difference in training dynamics between local training and backpropagation still limits such approaches (Fournier et al., 2023; Wang et al., 2021).

**Pipeline parallelism.** Pipelining encompasses a range of model parallel techniques that divide the components of a network into stages that compute in parallel, while avoiding idle workers. Initially popularized by Huang et al. (2019), a batch of data is divided into micro-batches that are processed independently at each stage. Although more efficient pipelining schedules have been proposed (Fan et al., 2021), notably to mitigate the peak memory overhead, keeping an exact batch gradient computation requires leaving a bubble of idle workers. By alternating one forward and one backward pass for each worker, PipeDream (Narayanan et al., 2019) can allow to get rid of idleness bubbles, but at the expense of introducing staleness in the gradients used. Narayanan et al. (2021) mitigates this staleness to only one optimization step by accumulating gradients, thus also reducing the parameter memory overhead to only two versions of the parameters. Nevertheless, these approaches still suffer from a quadratic activation memory overhead with regard to the number of stages, as micro-batch activations pile up in buffers, especially for early layers. Some implementations propose to limit this overhead by combining activation checkpointing (Chen et al., 2016) with pipelining (Kim et al., 2020; Liu et al., 2023), although the memory overhead still scales with the number of stages.

**Delayed gradient.** By allowing stale gradients in the update process, these previous methods provide the context for our approach. Delayed gradient optimization methods are model parallel techniques that aim to decouple and process layers in parallel during backpropagation. In these approaches, delays occur stage-wise: the backward pass may be computed with outdated parameters or activations compared to the forward pass. For instance, Huo et al. (2018a) proposes a feature replay approach, where a forward pass first stores intermediary activations, which are then "replayed" to compute the backward pass in parallel. This method still requires heavy synchronization between layers, yielding a lock on computations. In Zhuang et al. (2020) and Zhuang et al. (2021a), stale gradients are computed from older parameter versions differing from the parameters used during the update. This staleness can be mitigated: Zhuang et al. (2021a) 'shrinks' the gradient by the delay value, but more advanced techniques also exist (Yang et al., 2021; Kosson et al., 2021). Still, these methods are limited like previous pipelining methods by their memory overhead as the computational graph is fully stored. A first step to reduce this, as proposed in Diversely Stale Parameters (DSP) (Xu et al., 2019), PipeMare (Yang et al., 2021) and (Kosson et al., 2021), is to keep a single set of parameters and approximate the gradients computed during the backward pass with the updated parameters, which differ from the ones used in the forward pass. This requires, like in activation

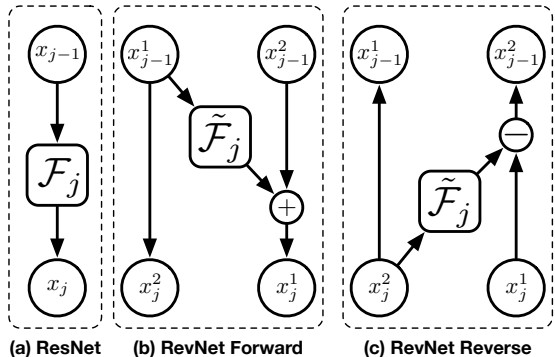

Figure 2: **Differences between the residual block of a ResNet and its reversible counterpart. (a)** Forward of a residual block. **(b)** Forward and **(c)** Reverse forward of a reversible residual block. For reversible blocks, as in Gomez et al. (2017), the input $x_j$ is doubled in size and split equally into $\{x_j^1, x_j^2\}$ along its channels. The function $\mathcal{F}_j$ includes a skip-connection while $\tilde{\mathcal{F}}_j$ does not.

checkpointing, an additional reconstruction of the computational graph. Furthermore, the quadratic activation memory overhead still limits the scalability of these methods for a large number of stages.

## 3 METHOD

### 3.1 STANDARD BACKPROPAGATION

We consider a DNN composed of $J$ stages (e.g., a layer or a set of layers). An input $x_0$ is propagated through the network, recursively defined by

$$x_j \triangleq \mathcal{F}_j(x_{j-1}, \theta_j),\tag{1}$$

where $\mathcal{F}_j$ is the $j$-th stage parameterized by $\theta_j$. The backpropagation algorithm is the ubiquitous algorithm to compute parameter gradients. First, an input is propagated through the network with a forward pass, while storing its intermediate activations. A scalar loss $\mathcal{L}$ is then deduced from the corresponding output $x_J$. Parameter gradients are then computed during the backward pass by taking advantage of the chain rule: starting from the last stage with $\delta_J = \nabla_{x_J}\mathcal{L}$, the gradients with regard to the activations are given by

$$\delta_j \triangleq \nabla_{x_{j-1}}\mathcal{L} = \partial_x \mathcal{F}_j(x_{j-1}, \theta_j)^{\mathrm{T}}\delta_{j+1},\tag{2}$$

and the gradients with regard to the parameters are defined as

$$\Delta_j \triangleq \nabla_{\theta_j}\mathcal{L} = \partial_\theta \mathcal{F}_j(x_{j-1}, \theta_j)^{\mathrm{T}}\delta_{j+1}.\tag{3}$$

Note that these computations follow a synchronous and sequential order. The parameters $\theta_j$ can then be updated given their gradient estimate $\Delta_j$, using any optimizer.

### 3.2 REVERSIBLE ARCHITECTURES

We focus on the reversible neural networks presented in Gomez et al. (2017), although our method is not dependent on this architecture. Note that this is a weak restriction as many architectures are adaptable to reversible ones Mangalam et al. (2022). In practice, only a few stages which do not preserve feature dimensionality are not reversible and correspond to the downsampling blocks in the ResNet. Fig. 2 highlights how reversible residual blocks $\mathcal{F}_j$ differ from their standard counterpart. The input is split into two equal-size inputs, along the channel dimension, that are propagated forward according to Fig. 2b using an ad-hoc operator $\tilde{\mathcal{F}}_j$. It can be reconstructed by reverse propagating the output according to Fig. 2c, by subtracting the output of $\tilde{\mathcal{F}}_j$ rather than adding it like in the previous forward.

Table 1: **Comparisons with other methods in an ideal setting for one stage.** We compare several methods to compute a gradient estimate in a model parallel setting: classical backpropagation, its reversible counterpart (Gomez et al., 2017), the Delayed gradients approach of Zhuang et al. (2020) and its improvements using checkpointing by Xu et al. (2019), and our proposed approach. Here, $J$ is the total number of stages while $j$ is the stage index. For the sake of simplicity, we assume that a backward pass requires approximately 2 times more FLOPs than a forward pass. *Full Graph* indicates that it is required to store the full computational graph of a local forward pass. With a limited increase in communication volume and FLOPs, PETRA requires the least storage of all methods while being *linearly* faster than backpropagation. We assume that the forward and backward passes can be executed in parallel for PETRA or delayed gradients, making the backward pass responsible for most of the computation time in parallelizable approaches.

| | Storage | | Comm. | FLOPs | Mean time |
| Methods | Activations | Params. | Volume | | per batch |
| --- | --- | --- | --- | --- | --- |
| **Backpropagation** | Full Graph (FG) | 1 | 1 | $\mathbf{3J}$ | $3J$ |
| **Reversible backprop.** | 0 | 1 | 4 | $4J$ | $4J$ |
| **Delayed gradients** | $2(J-j) \times$ FG | $\frac{2(J-j)}{k}$ | 1 | $\mathbf{3J}$ | **2** |
| **+ Checkpointing** | $2(J-j)$ | 1 | 1 | $4J$ | 3 |
| **PETRA (ours)** | 0 | 1 | 4 | $4J$ | 3 |

**Reversible stages.** In order to compute the exact gradients during the backpropagation phase, each reversible stage needs to retrieve its output from the stage above. We note $\mathcal{F}_j^{-1}$ the reverse stage function, which reconstructs the input from the output. We recursively apply the reconstruction to the final activation $x_J$, such that

$$\begin{bmatrix} x_{j-1} \\ \delta_j \end{bmatrix} = \begin{bmatrix} \mathcal{F}_j^{-1}(x_j, \theta_j) \\ \partial_x \mathcal{F}_j(\mathcal{F}_j^{-1}(x_j, \theta_j), \theta_j)^{\mathrm{T}} \delta_{j+1} \end{bmatrix} . \tag{4}$$

Note that reconstructing the input in our procedure is computationally equivalent to recomputing the activations in activation checkpointing, meaning it is equivalent to a single forward pass. Thus, this augmented backward procedure is equivalent to one regular forward call and backward call. However, one should observe that since the input $x_{j-1}$ must be sent to the reversible stages, this doubles the cost of backward communications.

**Non-reversible stages.** In practice, a reversible architecture includes layers that reduce dimensionality for computational efficiency, which thus correspond to non-invertible functions. For those very few stages, we employ a buffer mechanism to store activations and, like activation checkpointing, we recompute the computational graph with a forward pass during the backward pass. Note that this would not be the case when using invertible (i.e., bijective) architectures (Jacobsen et al., 2018a), which use an invertible downsampling.

### 3.3 A PARALLELIZABLE APPROACH: PETRA

As with any model parallel training technique, PETRA requires to partition the network architecture into stages $\mathcal{F}_j$ that are distributed across distinct devices. Each device $j$ needs only to communicate with its neighboring devices $j-1$ and $j+1$. The pseudo-code in Alg. 1 details the operations performed by each device, and the whole algorithm execution can be summarized as follows. The first device sequentially accesses mini-batches, initiating the data propagation process. When receiving its input $x_{j-1}^t$ from the previous stage, each stage processes it in forward mode and passes it to the next stage, until the final stage is reached. The final stage evaluates the loss and computes the gradients with regard to its input and parameters, thus initiating the backward process, which is performed in parallel of the forward process. In it, each stage processes the input and its associated gradient from the next stage. This means first reconstructing the computational graph, either while reconstructing the input $\tilde{x}_{j-1}^t$ for reversible stages or with a forward pass as in activation check-

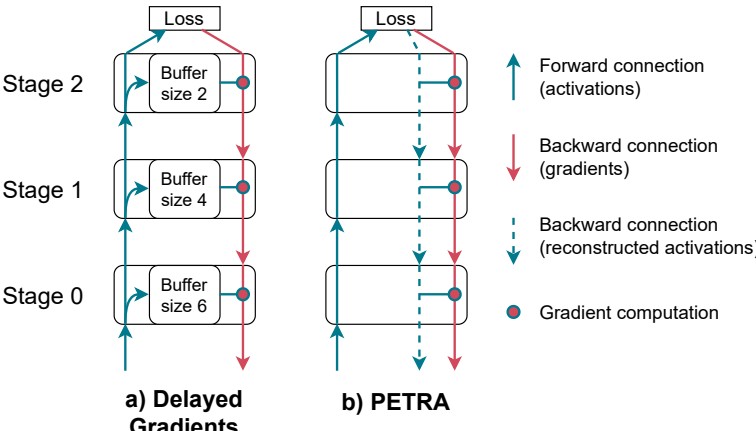

Figure 3: **Comparison of memory use between PETRA and a standard Delayed Gradient method** (Zhuang et al., 2020). By avoiding weight stashing and reversing the output into the input during the backward phase, we are able to fully decouple the forward and backward phases in all reversible stages, with no memory overhead, compared to standard delayed gradient approaches.

pointing otherwise. Then, the parameter gradient approximation $\Delta_j^{t+1}$ and the input gradient are computed before passing the latter to the previous stage. For intermediary reversible stages, this translates into the following equations, where $t$ corresponds to the current time step of the training,

$$
\begin{cases}
x_j^{t+1} = \mathcal{F}_j(x_{j-1}^t, \theta_j^t) \\
\tilde{x}_{j-1}^{t+1} = \mathcal{F}_j^{-1}(\tilde{x}_j^t, \theta_j^t) \\
\delta_j^{t+1} = \partial_x \mathcal{F}_j(\tilde{x}_{j-1}^{t+1}, \theta_j^t)^{\mathrm{T}} \delta_{j+1}^t \\
\Delta_j^{t+1} = \partial_\theta \mathcal{F}_j(\tilde{x}_{j-1}^{t+1}, \theta_j^t)^{\mathrm{T}} \delta_{j+1}^t \\
\theta_j^{t+1} = \mathrm{Optimizer}_j^t(\theta_j^t, \Delta_j^{t+1}).
\end{cases}
\tag{5}
$$

Note that this complete set of equations effectively decouples communications, computations, and parameter updates between independent devices. Indeed, reversible stages are able to operate without maintaining any state between the forward and corresponding backward phase by simply avoiding weight stashing, similarly to Xu et al. (2019), and by reversing the output into the input during the backward phase, removing the need for an input buffer. As parameters are updated between the forward and backward phases, the reversible stage produces an approximate input reconstruction, thus evaluating gradients with an approximate set of inputs and parameters during the backward phase. We illustrate in Fig. 3 the mechanism of PETRA compared to standard delayed gradient approaches that rely on additional buffers (Zhuang et al., 2021b; 2020).

**Complexity analysis.** We now discuss the benefits of our method, which are summarized in Tab. 1. In this discussion, we assume a homogeneous setting in which almost identical stages are distributed across $J$ devices uniformly. First, we consider the backpropagation setting, assuming a model parallelism strategy: a standard backpropagation pass requires storing locally both the parameters and the computational graph and due to the update lock of backpropagation (Jaderberg et al., 2017), requires synchronization between subsequent layers which impede the speed of computations. Standard Delayed Gradients strategies as implemented in Zhuang et al. (2021b; 2020) allow to unlock this barrier, but they require buffers for storing both the computational graph and parameters which can become impractical when using large models. In Xu et al. (2019), an activation checkpointing strategy removes the need for storing parameters, yet it requires a small computational overhead of 33% (assuming a backward pass is approximatively two times slower than a forward pass, see Fig. 6 of Huo et al. (2018b) and Mizutani & Dreyfus (2001)). To avoid storing activations, we rely on reversible architectures (Gomez et al., 2017) which increases the amount of forward communications by a factor of 2 and backward communication by a factor of 4 – activations sizes

---

**Algorithm 1** Worker perspective for training in parallel with PETRA, on a stage $j$, assuming initialized parameters $\theta_j$ and time step $t$, as well as an accumulation factor $k > 1$.

---

 1: ***In parallel*** *on the $j$-th stage,* $1 \leq j < J$, ***perform:***
 2:     **Forward Communications and Computations:**
 3:       **If** $j = 1$ **then**
 4:         $x_0 \leftarrow$ **Read**$_{\text{dataset}}$
 5:       **Else**
 6:         $x_{j-1} \leftarrow$ **Wait and Receive** $_{\text{from } j-1}$
 7:       **If** stage $j$ is not reversible :
 8:         **Buffer**$_j \leftarrow x_j$
 9:       $x_j \leftarrow \mathcal{F}_j(x_{j-1}, \theta_j)$
10:       **Send** $_{\text{to } j+1}(x_j)$
11:     **Backward Communications and Computations:**
12:       $(\tilde{x}_j, \delta_{j+1}) \leftarrow$ **Wait and Receive** $_{\text{from } j+1}$
13:       **If** stage $j$ is reversible:
14:         $\tilde{x}_{j-1} \leftarrow F_j^{-1}(\tilde{x}_j, \theta_j)$ *and keep computational graph in memory*
15:       **Else :**
16:         $\tilde{x}_{j-1} \leftarrow$ **Buffer**$_j$
17:         $x_j \leftarrow \mathcal{F}_j(\tilde{x}_{j-1}, \theta_j)$ *to recompute the computational graph*
18:       $\delta_j \leftarrow \partial_x \mathcal{F}_j(\tilde{x}_{j-1}, \theta_j)^T \delta_{j+1}$
19:       $\Delta_j \leftarrow \Delta_j + \frac{1}{k} \partial_\theta \mathcal{F}_j(\tilde{x}_{j-1}, \theta_j)^T \delta_{j+1}$
20:       **If** $t \bmod k = 0$ **then**:
21:         **Update** parameters $\theta_j$ with $\Delta_j$
22:         $\Delta_j \leftarrow 0$
23:       $t \leftarrow t + 1$
24:       **Send** $_{\text{to } j-1}(x_j, \delta_j)$
25:
26: ***In parallel*** *on the final stage $J$, perform:*
27:       $x_{J-1} \leftarrow$ **Wait and Receive** $_{\text{from } J-1}$
28:       $\mathcal{L} \leftarrow \mathcal{F}_J(x_{J-1}, \theta_J)$
29:       $\delta_J \leftarrow \nabla_{x_J} \mathcal{L}$
30:       $\Delta_J \leftarrow \Delta_J + \frac{1}{k} \nabla_{\theta_J} \mathcal{L}$
31:       **If** $t \bmod k = 0$ **then**:
32:         **Update** parameters $\theta_J$ with $\Delta_J$
33:         $\Delta_J \leftarrow 0$
34:       $t \leftarrow t + 1$
35:       **Send** $_{\text{to } J-1}(x_{J-1}, \delta_J)$

---

double and one has to pass both activations and gradients at the same time during backward. None of the aforementioned methods scale with the depth $J$: PETRA combines all the advantages of the previous methods, allowing an efficient parallelization scaling linearly with no memory overhead, while leading to a limited and constant overhead in computations and communications.

## 4 NUMERICAL EXPERIMENTS

### 4.1 CLASSIFICATION ACCURACY

We now describe our experimental setup on CIFAR-10 (Krizhevsky, 2009), ImageNet-32 (Chrabaszcz et al., 2017), and ImageNet (Deng et al., 2009).

**Experimental setup.** All our experiments use a standard SGD optimizer with a Nesterov momentum factor of 0.9. We train all models for 300 epochs on CIFAR-10 and 90 epochs on ImageNet32 and ImageNet. We apply standard data augmentation, including horizontal flip, random cropping, and standard normalization but we do not follow the more involved training settings of Wightman et al. (2021), which potentially leads to higher accuracy. We perform a warm-up of 5 epochs where the learning rate linearly increases from 0 to 0.1, following Goyal et al. (2017). Then, the learning

Table 2: **Classification accuracies using our PETRA method with RevNets, compared to standard backpropagation on ResNets and RevNets** on CIFAR-10, ImageNet32, and ImageNet. Our method delivers competitive results with backpropagation, even on ImageNet.

| Method | Model | Param. count | CIFAR-10 | ImNet32 | ImNet |
|---|---|---|---|---|---|
| Backprop | ResNet18 (PyTorch) | 11.7M | - | - | 69.8 |
| Backprop | ResNet18 (Ours) | 11.7M | 95.0 | 54.0 | 70.8 |
| Backprop | **Rev**Net18 (Ours) | 12.2M | 94.9 | 54.6 | 70.8 |
| PETRA | **Rev**Net18 (Ours) | 12.2M | 94.9 | 54.6 | 71.0 |
| Backprop | ResNet34 (PyTorch) | 21.8M | - | - | 73.3 |
| Backprop | ResNet34 (Ours) | 21.8M | 95.5 | 56.5 | 74.0 |
| Backprop | **Rev**Net34 (Ours) | 22.3M | 95.3 | 56.4 | 73.2 |
| PETRA | **Rev**Net34 (Ours) | 22.3M | 94.8 | 56.1 | 73.5 |
| Backprop | ResNet50 (PyTorch) | 25.6M | - | - | 76.1 |
| Backprop | ResNet50 (Ours) | 25.6M | 94.8 | 58.8 | 75.6 |
| Backprop | **Rev**Net50 (Ours) | 30.4M | 95.2 | 59.7 | 75.4 |
| PETRA | **Rev**Net50 (Ours) | 30.4M | 94.5 | 59.6 | 74.8 |

rate is decayed by a factor of 0.1 at epochs 30, 60, and 80 for ImageNet32 and ImageNet – it is decayed at epochs 150 and 225 for CIFAR-10. We use a weight decay of 5e-4 for CIFAR-10 and 1e-4 for ImageNet32 and ImageNet. As suggested in Goyal et al. (2017), we do not apply weight decay on the batch norm learnable parameters and biases of affine and convolutional layers. For our standard backpropagation experiments, we follow the standard practice and use a batch size of 128 on ImageNet32 and CIFAR-10, and 256 on ImageNet32. However, we made a few adaptations to train our models with PETRA. As suggested by Zhuang et al. (2020; 2021a), we employ an accumulation factor $k$ and a batch size of 64, which allows to reduce the effective staleness during training: in this case, $k$ batches of data must be successively processed before updating the parameters of a stage (see Alg. 1). Such gradient accumulation however also increases the effective batch size, and we apply the training recipe used in Goyal et al. (2017) to adjust the learning rate; note that we use the average of the accumulated gradients instead of the sum. The base learning rate is thus given by the formula $\mathtt{lr} = 0.1\frac{64k}{256}$, with $k$ the accumulation factor.

**Model adaptations.** For designing our RevNet architectures, we adopt a methodology similar to Gomez et al. (2017): the number of channels in each stage is multiplied by 2 to account for the second data stream according to Fig. 2. However, as the stage function $\tilde{\mathcal{F}}_j$ operates only on one of the two streams, the number of parameters stays almost the same between a residual block and its revertible counterpart. Consequently, the DNNs are split to preserve each residual block, resulting in 10 stages for RevNet18, and 18 stages for RevNet34 and RevNet50; thus varying the level of staleness between configurations. On CIFAR-10, the input layer uses 3x3 convolutions instead of 7x7 convolutions and does not perform max-pooling. The running statistics of batch normalization layers are updated when recomputing the activations during the backward pass and are then used during model evaluation – the running statistics are not updated during the forward pass.

**Performance comparison.** Tab. 2 reports our numerical accuracy on several vision datasets, comparing a backpropagation performance from an official PyTorch implementation of ResNets (the numbers can be found as v1 of `https://pytorch.org/hub/pytorch_vision_resnet/`), for our own implementation of ResNets and RevNets in our custom computational framework, and our proposed method, PETRA. For PETRA, we report the best classification accuracy after the last learning rate drop, using the best value (picked on the training set) of accumulation steps within $\{1, 2, 4, 8, 16, 32\}$. Our CIFAR-10 accuracies are averaged over 3 runs, with a variance smaller than 0.1. We observe that while our reversible models have about the same parameter count, they all perform in the same range of accuracy as their non-reversible counterparts. Only the RevNet-50 leads to a small drop in accuracy on ImageNet of about 0.6%: using different downsampling layers removes this gap at the expense of a substantial increase in the parameter count (30.4M to 50M). For the stake of comparison with the original ResNets, we did not include this result.

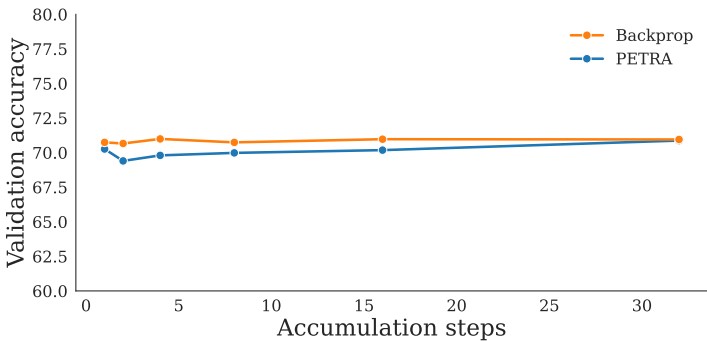

Figure 4: **Validation accuracy of PETRA and backpropagation for a various number of accumulation steps**, for a RevNet18 trained on ImageNet with $k \in \{1, 2, 4, 8, 16, 32\}$. The validation accuracies are averaged over the last 10 epochs. As the number of accumulation steps increases, the effective staleness in PETRA decreases, closing the gap with standard backpropagation.

Table 3: **Memory savings for RevNet50 on ImageNet with our method for different configurations.** We indicate the use of memory buffers for inputs or parameters. The savings are computed with respect to the first configuration, where inputs and buffers are stored. Our method achieves 54.3% memory reduction over the base configuration of Delayed Gradients.

| Buffer | | Memory (GB) | Saving (%) |
|---|---|---|---|
| **Input** | **Params.** | | |
| $\checkmark$ | $\checkmark$ | 44.5 | 0.0 |
| $\checkmark$ | $\times$ | 43.6 | 2.0 |
| $\times$ | $\checkmark$ | 21.2 | 52.3 |
| $\times$ | $\times$ | **20.3** | **54.3** |

**Impact of the accumulation $k$.** We test the impact of the accumulation on a RevNet-18 trained via PETRA for various values of accumulations $k \in \{1, 2, 4, 8, 16, 32\}$ on the ImageNet dataset. Fig. 4 indicates that our method can benefit from large accumulation factors, with the well-known trade-off of large batches mentioned in Goyal et al. (2017). Increasing the accumulation factor reduces the effective staleness during training, and closes the performance gap with standard backpropagation with perfect matching for $k = 32$. This confirms that this large-batch training recipe derived for synchronous data parallelism is also particularly suited for our model parallel approach.

## 4.2 TECHNICAL DETAILS

**A note on the implementation.** We shortly describe our implementation details. We base our method on PyTorch (Ansel et al., 2024), although we require significant modifications to the Autograd framework in order to manage delayed first-order quantities consistently with PETRA. We rely heavily on the *Vector Jacobian Product* of PyTorch to compute gradients during the backward pass of each stage, but other backends could be used. The backward pass for reversible stages only necessitates a reconstruction step and a backward step – a naive implementation would use a reconstruction step, followed by a forward and a backward step. This is because we only need the output gradient as well as the computational graph of $\tilde{\mathcal{F}}_j$ to compute the input and parameter gradients at line 12 and 13 of Alg. 1, which can be obtained during the input reconstruction phase. For non-reversible stages, we reconstruct the computational graph with a forward pass on the input retrieved from the buffer during the backward pass. Our models can run on a single A100, 80GB to easily compare training dynamics, or distributed over 10 or 18 GPUs when training with a RevNet-18 or a RevNet-34 or 50.

**Memory benefits and training time.** Here, we discuss the practical memory savings and throughput speedup. To better understand the advantage of our method compared to other delayed gradient

Table 4: **Performance on CIFAR-100 with our method for different configurations,** for different RevNets. Here, we use no accumulation ($k = 1$) to better pinpoint the effect of the staleness. We indicate the use of memory buffers for inputs or parameters and whether using exact reversible backpropagation or using delayed gradients. We note that using the loss of both of the buffers does not have a particular effect on performance compared to the delay, further justifying our approach.

| Delayed gradients | Buffer | | Accuracy | | |
|:---:|:---:|:---:|:---:|:---:|:---:|
| | Input | Params. | RevNet-18 | RevNet-34 | RevNet-50 |
| $\times$ | $\checkmark$ | $\checkmark$ | 77.5 | 78.1 | 78.5 |
| $\checkmark$ | $\checkmark$ | $\checkmark$ | 76.5 | 76.3 | 77.7 |
| $\checkmark$ | $\checkmark$ | $\times$ | 76.4 | 75.5 | 76.7 |
| $\checkmark$ | $\times$ | $\checkmark$ | 75.4 | 76.4 | 78.3 |
| $\checkmark$ | $\times$ | $\times$ | 75.9 | 75.2 | 77.2 |

Table 5: **Mean iteration time** of one micro-batch of size 256 of CIFAR-10 data, when training with PETRA or Reversible backpropagation. Training is distributed in both cases on 10 (for the RevNet-18) or 18 GPUs (for the RevNet-34/50). We observe a significant speed-up for the 3 models.

| Model | Rev. backprop. | PETRA | Speed-up |
|:---|:---|:---|:---|
| **RevNet-18** | 160.3ms | 53.4ms | 3.0$\times$ |
| **RevNet-34** | 235.6ms | 97.1ms | 2.4$\times$ |

approaches (Harlap et al., 2018; Xu et al., 2019; Kosson et al., 2021), we emphasize the practical memory savings associated with different methods in Tab. 3. We estimate the memory needed in gigabytes, as the sum of the size of the model, the input and parameter buffers (excluding the input buffer of the first stage, which uses retrievable dataset inputs). Here we do not include the effect of gradient accumulation, which depends on $k$ and would only affect the parameter buffer size, which is small in our case. Note that the batch size also affects the memory savings, and we set it to 64 for consistency with Tab. 2. Storing both inputs and parameters into a buffer corresponds to the PipeDream approach (Harlap et al., 2018), while only storing inputs would correspond to the approach in Xu et al. (2019); Kosson et al. (2021). Not storing inputs (lines 3 and 4) is only applicable to reversible architectures. The input buffer has the biggest impact on the total memory, being responsible for 52.3% of the memory footprint. Dropping the parameter buffer with PETRA pushes the savings further up to 54.3% for a RevNet50 on ImageNet. We report on-device stage memory in Tab. 6, where we note that non-reversible stages account for the majority of total memory use, indicating that savings would be much higher when using fully invertible architectures.

We also measure the effective throughput of training with PETRA a ResNet-18 on 10 GPUs and a ResNet-34 on 18 GPUs, and compare it with basic model parallelism, where batch computations are not overlapped between stages. After letting a warm-up period of 500 iterations, we report the effective throughput by measuring the processing time of 50 mini-batches and averaging it. As can be seen in Tab. 5, our method achieves wall-clock time speedup compared to basic model parallelism. While our main objective is to show the empirical effectiveness of our training procedure for convergence, we also observe a significant speed-up, despite using relatively unbalanced stages. A more efficient implementation in the future will allow further training speed improvements.

**Impact of the buffers on gradient estimation**   Three different approximations in gradient estimation are used in PETRA which may affect our performances, and we investigate in Tab. 4 their impact on CIFAR-100. The use of a gradient computed on previous iterations of the parameters is necessary to speed-up computations. This has the most impact on performances (line 2), but note that using accumulation vanishes this performance drop as shown in Fig. 4. Line 4 should corresponds to reversible backpropagation with delays, which is equivalent to line 2, but we choose to reconstruct the input using the latest parameters in the memory buffer, to better characterize the impact of reconstructing inputs with the latest parameters, as done in PETRA. Lines 4 and 5 indicate that approximating the input affects the learning dynamics more than approximating the weights.

## 5    CONCLUSION

In this work, we introduce PETRA, a novel model parallel training technique for reversible architectures which is a novel promising alternative to backpropagation. It achieves a significant parallelization with a limited overhead compared to standard backpropagation or other competitive alternatives to end-to-end training, like delayed gradients approaches. Our method has the potential to achieve linear speedup compared to standard backpropagation and allows reversible layers to operate without any parameter or activation buffers, effectively decoupling the forward and backward phases. Despite using an approximate delayed gradient estimate, our method delivers competitive performances compared to standard backpropagation on standard computer vision datasets.

In future work, we aim to implement and optimize PETRA for Large Language Models (LLMs), with a first baseline being Reformers (Kitaev et al., 2020), invertible transformers that have been shown to scale. This will validate further PETRA's effectiveness and robustness on fully-reversible architectures, solidifying its potential as a cutting-edge training technique.

### ACKNOWLEDGMENTS

This work was supported by Project ANR-21-CE23-0030 ADONIS and PEPR IA on grant SHARP ANR-23-PEIA-0008, and Sorbonne Center for Artificial Intelligence (SCAI) of Sorbonne University (IDEX SUPER 11-IDEX-0004). EB acknowledges funding from FRQNT New Scholar and support for computation from CFI. SR acknowledges funding from European Union under GA no. 101135782 (MANOLO project). This work was granted access to the AI resources of IDRIS under the allocations 2023-A0151014526 made by GENCI.

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

Table 6: **Memory usage of each stage** when training with PETRA on CIFAR-10 with a batch size of 256. Stages with the same memory usage are grouped together for ease of read.

| **RevNet-18** | **Stage(s) index(es)** | 0 | 1-2 | 3 | 4 | 5 | 6 | 7 | 8 | 9 |
| | **Memory (GB)** | 1,15 | 1,01 | 2,67 | 0,52 | 1,13 | 0,31 | 0,57 | 0,25 | 0,11 |
| **RevNet-34** | **Stage(s) index(es)** | 0 | 1-4 | 5 | 6-8 | 9 | 10-12 | 13 | 14-16 | 17 |
| | **Memory (GB)** | 1,15 | 1,01 | 4,2 | 0,52 | 1,66 | 0,31 | 0,71 | 0,25 | 0,11 |

# A    MEMORY USAGE BY STAGE

In Tab. 6, we report the memory usage with PETRA of each stage of the networks trained. Note that some stages have a much higher memory footprint than others, which is due to the presence of non-reversible stages in RevNets, which requires activation buffers.

# B    QUALITY OF GRADIENT APPROXIMATION AND DELAYS IN DEPTH

**PETRA gradient approximations**    The PETRA optimization procedure estimates gradients with two approximations. First, PETRA estimates delayed gradients, with a delay $\tau_j = 2(J - j)$ for each block $j$ of a network partitioned into $J$ blocks. According to Eq. 5, for the $j$-th layer, this would mean that $\Delta_j^{t+1} = \partial_\theta \mathcal{F}_j(x_j^{t-\tau_j}, \theta_j^{t-\tau_j})$. The implicit underlying hypothesis is that the delayed gradients can serve as an approximation of the end-to-end gradient, or, more largely, as a descent direction. Note that this is the gradient computed by standard approaches like Zhuang et al. (2020). Then, PETRA makes further approximations on this delayed gradient. First, having no parameter buffers, each layer will only use the latest available in memory weights $\theta_j^t$, which will differ from those used in the forward pass $\theta_j^{t-\tau_j}$. Second, the input used for gradient computations is not stored in a buffer for reversible layers, but reconstructed. For both the input reconstruction and the Jacobian computations, the layers use the "up-to-date" parameters $\theta_j^t$.

**Framework**    To investigate the quality of these approximations empirically, we trained a RevNet18 on CIFAR-10, divided into 10 stages, while tracking approximation quality metrics for each layer throughout training. These metrics are computed 15 times per epoch and averaged at the end of the epoch. The optimization procedure is the same as the one used to obtain Tab. 2, without gradient accumulation to emphasize the impact of delay. Note that the network has non-reversible stages, and thus input buffers, at stages $\{3, 5, 7\}$; the first stage is not reversible but can retrieve its input from the dataset. Our model is trained via PETRA, and we compare, at given snapshots throughout the training, gradients following PETRA, standard delayed gradient approaches Zhuang et al. (2020), and *standard backprop gradients* at various depths.

To compare two different gradients, the two metrics we record are their cosine similarity and their norm ratio:
$$\text{Cos-Sim}(x, y) = \frac{\langle x, y \rangle}{||x|| \cdot ||y||}, \quad \text{Norm-Ratio}(x, y) = \frac{||x||}{||y||}$$

**Approximation with the standard delayed gradient**    We report these metrics throughout training in Fig. 5. We observe several tendencies. First, in Fig. 5a, we observe that the gradient computed by PETRA is indeed a good approximation of the standard delayed gradient, as predicted. The alignment improves during training, with a particular jump at the last learning rate drop. This is expected as the discrepancy between $\theta_j$ and $\theta_j^{t-\tau_j}$ becomes negligible as the model converges. Similarly, the alignment improves for later layers, where fewer delayed parameters are used, and with a smaller delay. The reconstruction error is also lower, although this error "resets" after each input buffer. Note also that the ratio of the norms of the gradients, in Fig. 5c, follows a similar trend, and stays consistently close to 1.

**Approximation with the end-to-end gradient**    Then, in Fig.5b, we also compare the alignment of the PETRA gradient with the end-to-end one and the alignment of the delayed gradient with the end-to-end one. These values are much lower for early layers, where the gradient computed

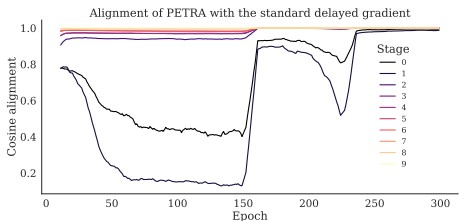

(a) Cosine similarity between PETRA and standard Delayed Gradients.

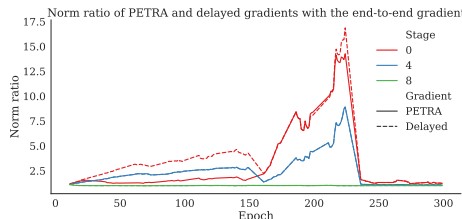

(b) Cosine similarities between PETRA and end-to-end gradients (full line), and between the standard delayed gradients (dashed line) and end-to-end ones.

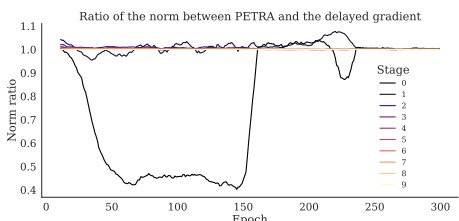

(c) Norm ratio between PETRA (numerator) and standard delayed gradients (denominator).

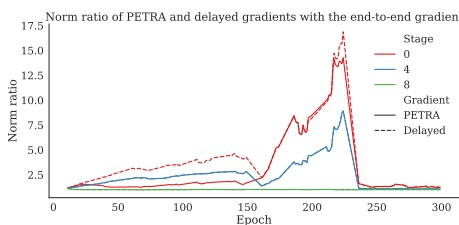

(d) Norm ratios between PETRA and end-to-end gradients (full line) and the standard delayed gradient and end-to-end gradient (dashed line).

Figure 5: Cosine similarities and norm ratios between gradients throughout training. Each point represents the average of 15 measurements during 1 epoch. Values are smoothed with a rolling window of size 10. Color corresponds to the stage index. The approximation is noticeably better after the last learning rate drop, and for later stages. Although PETRA approximates well the standard delay gradient, it also approximates better the end-to-end gradient compared to standard delay gradient approaches.

by both methods depends much more on the delay, and is thus less aligned with the end-to-end gradient. However, we surprisingly observe that the gradient computed by PETRA shows a better alignment with the end-to-end gradient compared to the delayed one. Although surprising, this can be explained by the fact that PETRA does not use delayed parameters during the backward pass, for both the input reconstruction and the Jacobian computations. Although reducing the alignment with the delayed gradient, this increases the alignment with the end-to-end one. The norm ratio between the delayed and PETRA gradients and the end-to-end ones is also quite high early in training, before coming close to 1 at convergence. Here again, the ratio is smaller for the PETRA gradient.

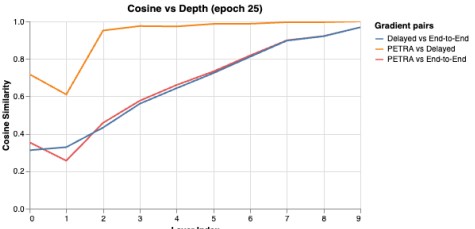

(a) Cosine similarities at epoch 25, i.e. 10 epoch after warm-up phase.

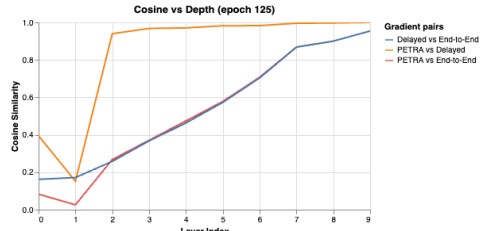

(b) Cosine similarities at epoch 125, i.e. 25 epoch before the first learning rate drop.

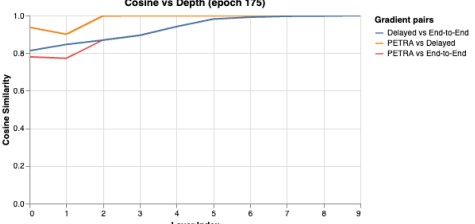

(c) Cosine similarities at epoch 175, i.e. 25 epoch after the first learning rate drop.

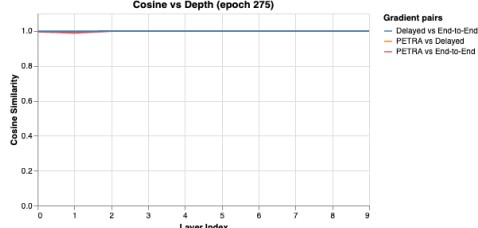

(d) Cosine similarities at epoch 125, i.e. 25 epoch after the second learning rate drop.

Figure 6: Cosine similarities against stage index of RevNet18 on CIFAR-10 at epochs 25, 125, 175 and 250. Approximations degrades between epoch 25 and 125, consistently with Fig. 5, but improves noticeably after each learning rate drop.

For a clearer comparison, we also provide in Fig. 6 cosine similarities values between the three gradients depending on the layer index, at different training epochs. We once again observe that the similarities improve as layer indexes increase and that the discrepancies between the gradients decrease across stages as the learning rate becomes smaller. We also observe that the similarity between PETRA and end-to-end is sometimes higher than for the delayed gradient, suggesting that using the up-to-date weights for the Jacobian computations and input reconstructions might help to mitigate the staleness.

