# OpenReview forum: "PETRA: Parallel End-to-end Training with Reversible Architectures"
_ICLR.cc/2025/Conference — ICLR 2025 Spotlight_

### Official Review · Reviewer_ggqx · 2024-11-03

**Soundness:** 3
**Presentation:** 4
**Contribution:** 3
**Rating:** 8
**Confidence:** 5

**Summary:**

This paper proposes to utilize the concept of reversible architectures to improve parallelization in DNN training. A model is split into multiple stages that are trained asynchronously; i.e. in a model parallel fashion. Leveraging reversibility, the training of the different stages is effectively decoupled. This scheme offers a linear speedup in the number of stages relative to end-to-end backprop, while reducing the memory footprint. The method is evaluated using ResNets/RevNets with three different image classification benchmarks.

**Strengths:**

The paper is well-written and easy to follow. The idea of utilizing reversibility for parallelization is a nice, simple, and novel idea! Consequently, I find myself sufficiently convinced that the method works --- albeit, that the empirical evaluation is somewhat limited. The novelty and applicability of the method mostly outweighs my concerns about the evaluation.

**Weaknesses:**

My only objection to this work is the limited number of experiments. They are limited to ResNet/Revnet 18/34/50 and CIFAR10, ImageNet-32, and ImageNet. It would definitely improve the paper to have at least a few more architectures included.

**Questions:**

How did you partition the architectures for your experiments? How many layers/blocks in each stage? Were they all the same size? And if so, would that not bring them out of sync during training such that top layers/stage were idle a lot of the time? The size of the feature maps is decreasing in the layer index, no? Thus, the lower layers/stages would consume more memory and compute than the top ones?

Perhaps you could add some information about this in the appendix :-)

---

> ### Author Response · Authors · 2024-11-23
>
> We thank the reviewer for his enthusiastic review and would like to address his questions.
>
> ### **How did you partition the architectures for your experiments? How many layers/blocks in each stage?**
>
> We partitioned the architectures at the residual block level, as standardly done for RevNets to obtain a fine-grained partitioning. The first convolutional block and the final projection each count as a separate stage; each residual block then also counts as a stage.
>
>
> ### **Were they all the same size? And if so, would that not bring them out of sync during training such that top layers/stage were idle a lot of the time?**
>
> We did not fully investigate idling time caused by the non-uniform distribution of FLOPS across stages. In this paper, we wanted to focus on the optimization properties of PETRA and the effect of crucial parameters. Therefore, we left the topic of fully optimizing resource efficiency for future work, which requires additional development efforts for automation. We also found it difficult to obtain a reasonable balance by hand, each configuration being hardware configuration specific. Ideally, we’d like every worker to have the same flops to fully exploit the parallelization potential. We plan to investigate this load balancing problem on transformers which maintains constant dimensionality and FLOPs across depth in future work.
>
>
> ### **The size of the feature maps is decreasing in the layer index, no? Thus, the lower layers/stages would consume more memory and compute than the top ones?**
>
> In order to properly answer this question, we need to separate compute from memory.
> - As the size of the feature maps is decreasing with respect to layer index, early layers consume more memory than the top ones.
> - The ResNet architectures are designed to approximately keep a constant number of FLOPs across layers. The non-reversible layers have more FLOPs than the other ones due to the extra downsampling operation, and the FLOPs of the different downsampling operations are not equal. The residual blocks that do not employ downsampling keep approximately the same number of flops, which is around 28 Giga-FLOPS for a ResNet-50.

---

### Official Review · Reviewer_md3p · 2024-11-03

**Soundness:** 3
**Presentation:** 3
**Contribution:** 4
**Rating:** 8
**Confidence:** 4

**Summary:**

The paper proposes to perform model parallel training using reversible architectures. Compared to delayed gradient, the proposed method is more memory efficient since it does not need to stash weights. It is shown that on shallower architecture the performance is slightly better than regular backprop and on deeper architecture such as ResNet-50, there is a slight drop but not significant. Overall, the work is likely to have a big impact as a way to scale up model parallel training.

**Strengths:**

- The paper demonstrated that activation reconstruction can work well with out-of-sync backward weights, and the reconstructed activations can be used to update weights.
- The paper has shown real computation and memory savings.

**Weaknesses:**

- It would be nice to see at what scale the method starts to break down (say when there is more and more delay in reconstruction). And show a plot on reconstruction error and final performance as a function of the number of delay steps. The model depth can be another variable to explore, aside from the few standard model architectures, perhaps sweeping a wider range of depths.
- Algorithm 1 is a little hard to process.
- The method relies on gradient accumulation to fully match with the It is unclear to me how gradient accumulation would have any impact when a large batch / data parallel is employed. This may not be a concern for LLMs, but for ImageNet and SSL training, many use very large batch sizes.

**Questions:**

N/A

---

> ### Author Response · Authors · 2024-11-23
>
> We thank the reviewer for his effort in this reviewing process, and would like to comment on the weaknesses reported.
>
> ### **It would be nice to see at what scale the method starts to break down (say when there is more and more delay in reconstruction). And show a plot on reconstruction error and final performance as a function of the number of delay steps. The model depth can be another variable to explore, aside from the few standard model architectures, perhaps sweeping a wider range of depths.**
>
> As the reviewer correctly noticed, the delay $\tau$ is the main source of numerical instabilities. Therefore, we chose to explore the largest setting up to a RevNet50, by partitioning the model at the residual block level. A coarser partitioning would decrease the delay and make the problem easier in our case.
>
> To address the concern of the reviewer, we designed a simple fully-reversible convolutional architecture by stacking reversible residual blocks. We train such architectures with PETRA while increasing the depth of the network and splitting the architecture at the residual block level, thus effectively increasing the delay during training. We will report the result in the rebuttal if they arrive in the following day, and include an appendix with the tables and plots suggested by the reviewer.
>
>
> ### **Algorithm 1 is a little hard to process.**
>
> We apologize for this. We deliberately chose a precise formulation of the algorithm to avoid any misinterpretation of the training process.
>
>
> ### **The method relies on gradient accumulation to fully match with the It is unclear to me how gradient accumulation would have any impact when a large batch / data parallel is employed. This may not be a concern for LLMs, but for ImageNet and SSL training, many use very large batch sizes.**
>
> We did investigate the effect of large batch sizes on ImageNet training. The available training recipes do not scale to effective batch sizes larger than 2048 without performance degradation; deterioration starts at 4096 in our experiments. Since LLMs are trained with significantly larger batch sizes, gradient accumulation would not be an issue in this scenario. It is indeed often used to handle significant model sizes when the maximum supported batch size on a given hardware configuration is too restricted. We however note that the training recipes allowing us to scale synchronous SGD to large batch sizes are also effective with PETRA according to our experiments.

---

> > ### Comment · Reviewer_md3p · 2024-11-24
> >
> > Thank you for the response. I look forward to your updated results.

---

### Official Review · Reviewer_hEA1 · 2024-11-04

**Soundness:** 2
**Presentation:** 3
**Contribution:** 2
**Rating:** 6
**Confidence:** 4

**Summary:**

PETRA is a model-parallel training method for reversible neural networks that decouples forward and backward passes, eliminating the need for activation or parameter buffers. This enables efficient parallel computation across devices with reduced memory overhead. PETRA matches backpropagation in accuracy on datasets like CIFAR-10 and ImageNet, while also achieving notable speed and memory savings, making it a potential alternative for large-model training.

**Strengths:**

The PETRA paper presents a new alternative for large-scale neural network training, offering efficient parallelization by decoupling forward and backward passes, which enables stages to compute independently across devices. Utilizing reversible architectures, PETRA removes the need for activation and parameter storage, achieving up to 54.3% memory savings, making it especially valuable for training large models. It demonstrates accuracy comparable with backpropagation on datasets like CIFAR-10 and ImageNet.

**Weaknesses:**

Dependency on Reversible Architectures: The approach is designed specifically for reversible architectures, which may limit its application to models that can be easily adapted to this structure. Non-reversible architectures, such as standard ResNets or some types of transformers, may not benefit as fully from PETRA’s memory and efficiency gains.
Increased Communication Overhead: While PETRA reduces memory usage, its reversible stages require additional communication overhead during the backward pass, which could affect scalability on very large, distributed systems. And the PETRA propose dividing a model into some
Scalability Constraints with Non-Reversible Layers: Although PETRA performs well on reversible architectures, any non-reversible stages still require stored activations, potentially increasing memory use and complicating scalability for models that include such layers.

**Questions:**

How PETRA perform on large model and more complex task, such as pretraining language model? The experiment in the paper is weak. The scalability of PETRA can not be verified by the current empirical results. Experiments on distributed pretraining for llm is necessary to validate the efficiency of PETRA, for example: experiments on Pile dataset with varying model size.

Is the reversible architecture necessary for PETRA? For models that integrate both reversible and non-reversible layers, how does PETRA manage memory savings and efficiency, and could these hybrid architectures affect its scalability benefits?

---

> ### Author Response · Authors · 2024-11-23
>
> We thank the reviewer for his dedicated time and feedback in this review process. We would like to comment on the reported weaknesses first:
> - **Dependency on reversible architectures:** the authors acknowledge that many existing architectures may not easily be made fully revertible. However, we believe that the favorable memory consumption of PETRA should motivate further research in developing fully reversible architectures.
> - **Increased Communication Overhead:** The communication overhead stems from the transition from a non-reversible to a reversible counterpart in our paper. While this is a valid concern when dealing with very large models, we want to emphasize that the increase is by a fixed factor, and does not depend on the network architecture. Therefore, we do not believe this would be a significant limiting factor of PETRA.
> - **Scalability Constraints with Non-Reversible Layers:** While this is true that non-reversible layers induced a memory overhead in PETRA, all other model parallel training techniques in the literature providing acceleration do require activation buffers and suffer from the same memory constraint. To the best of our knowledge, PETRA offers the best compromise in terms of memory consumption and acceleration.
>
> The authors will now attempt to answer each of the reviewer’s question:
> ### **How PETRA perform on large model and more complex task, such as pretraining language model?**
>
> The implementation used to compute the accuracy figures reported in the experiment section used a simulated environment to assess the numerical stability of PETRA, and did not allow us to perform efficient distributed training. Only recently did we succeed in producing an efficient distributed implementation of PETRA, from which we were able to derive runtime estimates to assess speedups for RevNets. While LLMs are not yet fully integrated into our framework for complete benchmarking, we have promising developments towards this use case with LLama2 on OpenWebText.
>
> Still, LLM pre-training requires substantial engineering to be performed efficiently. Thus, we consider that such large-scale studies are out of the scope of this paper. With the resources at our disposal, we focused our experience toward an academic-friendly setup as we personally think that the optimization properties must be studied in detail as we scale up the complexity of the task.
>
> [Edit]: One related concern, that another reviewer has raised explicitly, is the extent to which the depth, or more precisely the number of stages since it is related to the induced delay, affects the scalability potential of PETRA. To this end, we have currently performed experiments in Appendix B to ablate the effect of depth on gradient quality compared to backpropagation, and we have included it in the appendix of our current paper.
>
> ### **Is the reversible architecture necessary for PETRA?**
>
> PETRA does necessitate a reversible architecture. However, reversible architectures can be derived from many existing architectures, with similar optimization properties as shown in the literature; see [1, 2]. PETRA aims to democratize the development of reversible architectures by providing acceleration with limited memory overhead when scaling model size.
>
> ### **For models that integrate both reversible and non-reversible layers, how does PETRA manage memory savings and efficiency, and could these hybrid architectures affect its scalability benefits?**
>
> The presented paper proposes an implementation of PETRA able to handle non-reversible layers by using an activation buffer between the forward and backward pass. This induces a memory overhead similar to any other delayed gradient approach in the literature, which is quantified theoretically in the storage column of table 1. Resorting to buffers is necessary for our experiments which use the reversible counterpart of ResNets that include non shape-preserving layers which cannot be made invertible canonically as presented in the paper. We tried to specifically deal with the memory issue of non-reversible layers by deferring the buffer to the CPU memory. However, we did not succeed in obtaining an efficient implementation of this mechanism, and left this aspect for further optimization. This would require advanced profiling tools that we are trying to integrate within our code base. As a workaround, we also experimented with quantization as a way to decrease buffer size with promising results, but did not fully investigate the impact of such numerical approximation. We will add an experiment in the appendix quantifying the impact of buffer quantization to a lower precision on final model performances and report corresponding memory usage.
>
> [1] - Jacobsen, J. H., Smeulders, A., & Oyallon, E. (2018). i-revnet: Deep invertible networks. ICLR 2018.
>
> [2] - Kitaev, N., Kaiser, Ł., & Levskaya, A. (2020). Reformer: The efficient transformer. ICLR 2020.

---

> > ### Comment · Reviewer_hEA1 · 2024-11-30
> > **Response to rebuttal**
> >
> > I thank the authors for the thorough effort they put into their rebuttal. Some of my concerns were addressed, and I will update my scores accordingly.

---

### Official Review · Reviewer_A9EF · 2024-11-04

**Soundness:** 3
**Presentation:** 4
**Contribution:** 3
**Rating:** 8
**Confidence:** 4

**Summary:**

The authors propose a new algorithm for training reversible models. Compared to backpropagation, it can be run on each layer in parallel and with a reduced memory cost. They show empirically the advantages of their algorithm on RevNet models for image classification.

**Strengths:**

- The paper is well written, clear, and has helpful illustrations.
- The algorithm seems simple, natural and intuitive.
- While the algorithm relies on reversible layers, it can still be mixed with standard non-reversible layers, for which a standard backpropagation is performed.
- The authors validate their algorithm with thorough experiments and analyses.

**Weaknesses:**

1. Invertible networks are currently not very used. This limits the direct applications of the algorithm. However I am aware that PETRA could motivate such the use of such architectures.
2. The experiments are only performed on RevNet models for image classification. As mentioned in the conclusion, it would be very nice to see experiments on more tasks and models. Indeed, as PETRA is applicable to only a subset of models (reversible models), it is frustrating to only see experiments on a single architecture.
3. Lines 509-510: I think you meant RevNet instead of ResNet.

**Questions:**

- How is the approximated gradient influenced by the depth of the model? I would expect the error to increase as the model gets deeper.

I find the paper very interesting and am ready to increase my grade should my remarks be addressed by the authors.

---

> ### Author Response · Authors · 2024-11-23
> **Experiments for further insights**
>
> We thank the reviewer for his feedback and would like to add some elements to clarify his concerns. We highly appreciate that the reviewer acknowledges the scaling potential of reversible architecture within current memory constraints given suitable training procedures. We also acknowledge that experiments on other architectures would be very valuable, and are actively working on integrating transformers for both computer vision and NLP tasks. We also thank the reviewer for pointing out the typo at lines 509-510.
>
> We are currently running experiments to track metrics quantifying approximation quality throughout training. We should be able to report them within the rebuttal period, but are still waiting for the results to support our answer with empirical measurements. We are also running experiments with a fully reversible convolutional architecture where we are able to increase depth without memory overhead; albeit the architecture is intended to be a toy example, this will give some insights about PETRA’s performance on non-standard architectures. We will come back to the reviewer once all results have been gathered.

---

> ### Author Response · Authors · 2024-11-27
>
> We would like to thank the reviewer once again for their suggestion and take this opportunity to address a few points.
>
> ## Invertible networks are currently not very used. This limits the direct applications of the algorithm.
> Invertible networks are currently not widely used but they have been shown to be extendable to all major architectures without loss of accuracy. For example such as [1], [2], [3], and [4]. Each of these studies highlights that invertibility is an intrinsic property of the model that does not degrade performance. On the contrary, it enables reversibility through simple adaptations of the model, often resulting in improvements, computational savings, or enhanced interpretability.
>
> ## The experiments are only performed on RevNet models for image classification.
> The experiments in our work are conducted solely on RevNet models. We understand the reviewers' concerns; this paper serves primarily as a proof of concept, demonstrating that PETRA represents a promising research direction. In Appendix B, we have included additional experiments analyzing gradient approximation quality, emphasizing the potential for this work to extend beyond its initial scope, which we discuss below.
>
> ## How is the approximated gradient influenced by the depth of the model?
> We have proposed a detailed analysis of this in Appendix B. It demonstrates that, when comparing PETRA during training with standard backpropagation and delayed gradients (as used in [5]), PETRA produces consistent gradients. Notably, deeper layers exhibit gradients that are closer to the true end-to-end gradients, whereas delays introduce discrepancies. However, our experiments in Section 4 show that these discrepancies do not negatively impact training.
>
> **References**
>
> [1] - Kitaev, N., Kaiser, Ł., & Levskaya, A. (2020). Reformer: The efficient transformer. ICLR 2020.
>
> [2] - Gomez, A. N., Ren, M., Urtasun, R., & Grosse, R. B. (2017). The reversible residual network: Backpropagation without storing activations. NeurIPS 2017.
>
> [3] - Behrmann, J., Grathwohl, W., Chen, R. T., Duvenaud, D., & Jacobsen, J. H. (2019, May). Invertible residual networks. ICML 2019.
>
> [4] - Jacobsen, J. H., Smeulders, A., & Oyallon, E. (2018). i-revnet: Deep invertible networks. ICLR 2018.
>
> [5] Zhuang, H., Wang, Y., Liu, Q., Zhang, S., & Lin, Z. (2021). Fully Decoupled Neural Network Learning Using Delayed Gradients. IEEE transactions on neural networks and learning systems.

---

> ### Comment · Reviewer_A9EF · 2024-12-01
>
> I thank the authors for their rebuttal, which addressed my concerns. I am updating my score.

---

### Comment · Area_Chair_v4xE · 2024-11-24

Dear Reviewers,

This is a gentle reminder that the authors have submitted their rebuttal, and the discussion period will conclude on November 26th AoE. To ensure a constructive and meaningful discussion, we kindly ask that you review the rebuttal as soon as possible and verify if your questions and comments have been adequately addressed.

We greatly appreciate your time, effort, and thoughtful contributions to this process.

Best regards,
AC

---

### Comment · Area_Chair_v4xE · 2024-11-27

Dear Reviewers,

We wanted to let you know that the discussion period has been extended to December 2nd. If you haven't had the opportunity yet, we kindly encourage you to read the rebuttal as soon as possible and verify whether your questions and comments have been fully addressed.

We sincerely appreciate your time, effort, and thoughtful contributions to this process.

Best,

AC

---

### Meta-Review · Area_Chair_v4xE · 2024-12-16

**Metareview:**

The authors propose a new algorithm for training reversible models. Compared to backpropagation, it can be run on each layer in parallel and with a reduced memory cost. They show empirically the advantages of their algorithm on RevNet models for image classification.

**Additional Comments On Reviewer Discussion:**

All reviewers agree this is a good paper and it should be accepted.

---

### Decision · Program_Chairs · 2025-01-22

Accept (Spotlight)